# Evaluating the Potential Health Risks of Selected Heavy Metals across Four Wastewater Treatment Water Works in Durban, South Africa

**DOI:** 10.3390/toxics10060340

**Published:** 2022-06-20

**Authors:** Babatunde Femi Bakare, Gbadebo Clement Adeyinka

**Affiliations:** Environmental Pollution and Remediation Research Group, Department of Chemical Engineering, Mangosuthu University of Technology, Durban 4031, South Africa; bfemi@mut.ac.za

**Keywords:** heavy metal pollution, influent and effluent samples, non-carcinogenic, cancer risk

## Abstract

Poor and inadequate sanitation systems have been considered not only a human health issue, but also an environmental threat that instigates climate change. Nine heavy metals—arsenic (As), cadmium (Cd), cobalt (Co), chromium (Cr), iron (Fe), manganese (Mn), nickel (Ni), lead (Pb), and zinc (Zn)—were evaluated in influent and effluent water samples from four wastewater treatment plants in the Durban metropolis, KwaZulu-Natal, South Africa. The results indicate that the mean concentrations of all the heavy metals in the influent samples ranged from 0.122 to 1.808 mg/L, while the effluent samples had a concentration ranging from 0.118 to 0.854 mg/L. Iron was found to be in the highest concentration and the concentration of Co was the lowest across the wastewater treatment plants. The levels for most of the heavy metals in this study were found to be above the recommended maximum concentrations in surface and effluent waters as stipulated by the World Health Organization, United States Environmental Protection Agency, Food and Agriculture Organization, and the Department of Water Affairs and Forestry of South Africa. According to the toxicity effect due to non-carcinogenic risks, As, Pb, Cr, and Cd are considered to be of medium risk in this study, indicating that a probable adverse health risk is very likely to occur. Additionally, the cancer risk (RI) values were lower than 10^−3^, which shows that cancer development is very likely in individuals who are exposed. Cancer development associated with dermal absorption is quite negligible; thereby, it does not raise any concerns.

## 1. Introduction

Anthropogenic activities due to population growth have been a considerable source of pollution to surface water and groundwater globally. The expansion of urban areas, due to human migration from low population density areas in recent times, has witnessed an upward trend in most cities across the world. As a result, nearly all the existing facilities in cities are overstretched, with which the already planned structures cannot cope. To mitigate the spread of diseases and reduce environmental pollution, it is important that the waste generated by humans is safely handled and effectively treated. In developed countries, cities are planned in such a manner that the sewers systems are channeled appropriately into a wastewater treatment plant, which is capable of treating such waste. Unfortunately, population explosion and the poor management of sewer facilities have led to an uncontrollable situation where sewage enters the open space and, finally, finds its way into surface and underground waters. Approximately 4.2 billion people, which is estimated to be more than half of the world’s population, still lack access to safe sanitation regardless of sewer systems, which could lower the risk of contracting avoidable diseases and chronic illnesses [1]. Poor and inadequate sanitation systems are currently considered not only a human health issue, but also an environmental threat with a high potential to instigate climate change. It has been documented that nearly 80 percent of untreated wastewater from developed and developing countries enters diverse environments around the world [2]. Consequently, the excessive release of nutrients, such as nitrogen and phosphorous, into the environment from untreated wastewater could pollute natural ecosystems and disrupt aquatic life. The impact of raw sewage (wastewater) on pollution loads in the environment is significant, as most treatment plants receive their wastewater from either domestic, industrial, pharmaceutical, or agrochemical origins. Due to the pollution of the aquatic aquifers originating from the introduction of certain contaminants, many aquatic organisms have been brought to extinction [3]. Municipal, industrial, pharmaceutical, and agrochemical wastes carry various contaminants into surface water and groundwater systems, which serve as a primary route of pollution. Among these contaminants are organic pollutants, such as pesticides, industrial chemicals (e.g., polychlorinated biphenyls and dioxins), antimicrobial agents, over-the-counter medications, and heavy metals, which are toxic and have the potential to affect the physiological functions of living organisms and humans. Sewage effluent discharge has been associated with high levels of heavy metals, which may, in turn, pollute the receiving surface water [4]. Other sources of heavy metal pollution in the natural aquatic environment include leaching from the soil by acid rain, lead from exhaust vehicles and non-point sources, such as atmospheric deposition, precipitation, mining and refining operations, and the preparation of nuclear fuels [5,6,7]. Heavy metals are readily soluble in water or are adhered to the suspended particles in water and their concentration is in most cases unchanged even during degradation processes [8,9]. Heavy metal pollution in wastewater is a complex mixture that is hydrophilic and may remain in the water column for a long period of time, depending on the pH of the water environment. Heavy metals are recalcitrant in the environment, non-biodegradable, and toxic, and approximately half of them are released into the environment in quantities that are hazardous to the environment and impact human health negatively [10]. Studies have shown that conventional treatment methods for wastewater in most cases are not capable of removing many contaminants, such as heavy metals, effectively; thus, they are adsorbed into suspended particles in water and could easily be discharged with the effluents down in the water column and become a potential secondary source of pollution, thereby threatening ecosystems [3,11,12,13,14]. Although other remediation methods, such as adsorption, ion exchange, and phytoremediation processes, have been effective for heavy metal removal from aqueous solutions, Naushad et al. [15] conducted batch adsorption experiments using a TIV cation exchanger and reported that the method was effective for the removal of Pb^2+^ and Hg^2+^ metal ions from aqueous solutions. Similarly, a plant-based technology (phytoremediation) had been reported to clean contaminated lands [16], while ion exchange has also been reported to offer a potential removal of Cd^2+^, Co^2+^, Cu^2+^, and Pb^2+^ [17]. Research has shown that effluent samples contain substantial amounts of heavy metals. Bahiru [18] determined some levels of heavy metals in wastewater effluent samples and their toxicological implications in the Eastern Industrial Zone, Central Ethiopia. The mean concentrations of the selected heavy metals were in the range of 0 0.04–5.13 mg/kg. Khan et al. [19] reported significant amounts of heavy metals (Cr, Cu, Ni, Pb, Zn, and Cd) in wastewater-irrigated soils and food crops in Beijing, China. Due to the improper removal of heavy metals during the treatment process, waste effluents have been reported to release significant amounts of heavy metals. In this case, Arora et al. [20] reported a significant build-up of heavy metals in food crops due to the continuous use of wastewater for irrigation in India. The emergency of COVID-19 has recently changed the usual ways of life for human beings globally, whereby non-pharmaceutical approaches, such as the use of facial masks, have been adopted as a means of preventing the spread of coronavirus disease. In view of this, the environmental impact of the use of facial masks is a new aspect to the existing source of heavy metal pollution in surface water systems. In an attempt to assess the level of pollution associated with the heavy metal pollution originating from facial masks, Bussan et al. [21] quantified and determined the total concentration of heavy metals in various face masks using Inductively Coupled Plasma Mass Spectrometry (ICP-MS). The results revealed that most of the analyzed masks were found to contain trace elements below the detection limits, while few masks contained trace levels of heavy metals, such as Pb (13.33 mg/kg), Cu (410 mg/kg), Zn (56.80 mg/kg), and Sb (90.18 mg/kg) [21]. In recent times, the availability and supply of potable water in South Africa have witnessed a dramatic decline. The inadequate supply of safe water due to water stress conditions in the country has increased the demand for surface water, where people could be exposed to avoidable diseases and other related health issues and ecological imbalances due to pollution, which may arise from anthropogenic sources. Additionally, the recent emergence of the COVID-19 pandemic has not only affected people’s behavior, but its overall effects have contributed to creation of pollution sources, such as heavy metals, in the aquatic environment, which emanate from the non-pharmaceutical means of preventing the spread of the virus. The level of pollutants, such as heavy metals, in the raw sewage that frequently ends up in surface water requires constant and routine evaluation to safeguard the aquatic environment and the health of humans, which prompted this study. To curtail this problem, it is crucial to effectively remove heavy metals from sewage effluents as the excessive release of heavy metals into surface water and their accumulation in the soil may lead to a potential risk to human health as they are capable to undergo a transformation process and, in turn, bio-accumulate and be taken up by plants and enter the food chain. Therefore, the objectives of this study are: to investigate the levels of selected heavy metals, such as arsenic (As), cadmium (Cd), cobalt (Co), chromium (Cr), iron (Fe), manganese (Mn), nickel (Ni), lead (Pb), and zinc (Zn), which may pollute the environment if present beyond the recommended levels; and to evaluate their potential health risk and ecotoxicological effects, such as non-carcinogenic and cancer risks. This study is crucial as the effluents from these wastewater treatment plants are discharged directly into the surface rivers that pass through informal settlements where they are being used for various anthropogenic purposes, such as subsistence farming, washing, and bathing as well as drinking by open grazing animals. The cancer-related risk and health impacts of these heavy metals are important in the context of knowing the current state of the receiving rivers due to their potential to impair mental and neurological functions, influence neurotransmitter production and alter the various metabolic body processes of human beings, and the potential negative effects on aquatic lives.

## 2. Materials and Method

### 2.1. Description of the Study Location

The study areas were the Isipingo wastewater treatment work (29°59′24.97″ S, 30°54′21.81″ E), Southern wastewater treatment work (29°55′51.30″ S, 30°59′53.1204″ E), Northern wastewater treatment work (29°57′20.2176″ S, 30°59′52.098″ E), and New Germany wastewater treatment work, which is known as Innovative water treatment (29°45′24.264″ S, 30°51′44.856″ E). The global positioning system (GPS) was used to supply accurate sampling position sites. These treatment wastewater works are located across the Durban and Umlazi catchment areas under the eThekwini Metropolitan Municipality, KwaZulu-Natal, South Africa. The Southern wastewater treatment work (SWWTW) is located in Wentworth Valley, Bluff, and receives the majority of its raw sewage effluent through three large (1500 mm diameter) trunk sewers, i.e., Main Southern (“Jacobs”), Wentworth Valley, and Umlaas Trunk Sewers. Other smaller diameter pipelines that come to the plant include those from Mondi, SAPREF, and Illovo. The total average daily flow to this plant is ±130 ML/day. The Umlaas trunk sewer, serving Chatsworth and Umlazi, is predominantly domestic in origin, with a discharged flow of ±35 ML/day. This plant discharges all its treated flows directly to the sea through a 4.2 km long 1500 mm diameter sea outfall [22]. The Isipingo wastewater treatment work (WWTW) is located in the lower catchment Malukazi Malagazi, Umlazi. The facility was built in the late 1960s; it collects its raw sewage from the domestic communities within its catchment and discharges an average of 10.98 ML/day of treated effluent into the Isipingo River [23]. The Northern wastewater treatment plant (NWWTP) is located at 199 Johanna Rd, Peter Road, east Durban, with about 35 industries. The plant receives a capacity of about 60,000 L/day, with a monthly capacity of 18.27 ML/month [24], and discharges its treated effluent into the uMgeni River. The New Germany wastewater treatment work (NGWWT) is located at Unit 2 Devon Centre, Durban, 15 Devon Rd, New Germany. More details about the site locations and map are provided elsewhere by Bakare and Adeyinka [25].

### 2.2. Sample Collection and Preparation

The samples (raw influent and final effluent) were collected (three each at different points) across the four WWTPs, as discussed earlier. A composite sample was obtained by mixing equal water volumes of wastewater collected at regular time intervals. Sampling was performed in 15–21 September 2021. A 250 mL plastic bottle was used for the collection of the samples. The physicochemical parameters of the water samples, such as pH, conductivity temperature, and total dissolved solids (TDS), were measured onsite as a change in these parameters may alter the concentrations of the heavy metals in the samples. The integrity of the samples was maintained, and samples were transported safely to the laboratory at low temperatures using a cooler box with ice. Upon reaching the laboratory, the samples were filtered through joint-suction filtration glass Buchner funnel conical flask filters to remove suspended particulates. This is important to avoid the targeted heavy metals being adsorbed onto the suspended particulates in the raw samples; this was carefully performed so the volume of the sample was not altered for final dilution factor estimation. The filtered wastewater samples were preserved (acidified) using a 1.5 mL of HNO_3_ to minimize the precipitation and adsorption of heavy metals on the container walls [26] and stored in a refrigerator at 4 °C prior to further analyses at the Department of Chemical Engineering, Mangosuthu University of Technology, Umlazi. The filtered wastewater samples were digested using HNO_3_ heated on a block at 150 ± 20 °C as described by the EPA, Method 3050B (SW-846) (1996) [27]. The digested samples were filtered using 0.45 µm Acrodisc syringe filters, prior to analysis by inductively coupled plasma optical emission spectrometer (ICP-OES, PerkinElmer, Waltham, Massachusetts, United States) at a selected wavelength. The ICP-OES parameters were: argon gas was set at 415.82 kPa, power at 1.19 Kw, plasma at 9.99 and auxiliary at 0.60, and carrier flow at 0.70 with a low purge flow. The rotation pump was set at 20 revolutions per minute (20 r.p.m) with a speed temperature set at 37.99 °C, while the CCD temperature was operated at −14.99 °C and the vacuum level was at 1.5 Pa. The target heavy metals were As, Cd, Co, Cr, Fe, Mn, Ni, Pb, and Zn.

### 2.3. Quality Control

A record of every collected sample was made by carefully labelling each sample bottle with the cleared unique sample number. Each sample was collected beneath the surface of the sewage flow and an open sampling container was placed below and directed toward the sample current to avoid collecting surface scum. The mixing of grab samples to obtain composite samples and the acidification of filtered wastewater samples were carefully performed so that the dilution caused by mixing and acidifying samples was insignificant for a dilution correction factor purpose. Analytical grade HNO_3_ was used in this study to achieve the compatibility of the isolation of metals in the sample, which could aid an effective and direct determination after dilution and minimize the risk of environmental contamination. All the sample containers used in this study were shocked for two days with 10% (*v/v*) HNO_3_ and rinsed three times before use with Milli-Q ultra-pure water obtained from the Chemical Engineering Department, Mangosuthu University of Technology. Blank determinations were carried out for each set of analyses using the same reagents. All the sample preparations were performed timely to minimize the loss of heavy metals in the samples. The analyses were performed in duplicates. Instrument calibration using the selected heavy metals (a five-point calibration each) was performed together with all the samples to confirm the sensitivity status of the ICP-OES. The instrument response factor (R^2^) for each heavy metal was found to range from 0.998 to 0.999. The limits of detection and quantifications were evaluated using three and ten times the standard deviation of the blank with a slope of the regression line, respectively. The detection limits were 0.066 µg/mL (As), 0.0294 µg/mL (Cd), 0.0583 µg/mL (Co), 0.0281 µg/mL (Cr), 0.0595 µg/mL (Fe), 0.0220 µg/mL (Mn), 0.0473 µg/mL (Ni), 0.0249 µg/mL (Pb), and 0.0636 µg/mL (Zn). The limit of quantification was 0.219 µg/mL, 0.098 µg/mL, 0.194 µg/mL, 0.094 µg/mL, 0.198 µg/mL, 0.073 µg/mL, 0.157 µg/mL, 0.083 µg/mL, and 0.212 µg/mL for As, Cd, Co, Cr, Fe, Mn, Ni, Pb, and Zn, respectively. The recovery study was performed by spiking a known concentration of each heavy metal in deionized water, followed the same procedure as it was performed for the real samples. The percentage recovery was found to be 82.93%, 84.98%, 91.91%, 86.87%, 91.30%, 94.85%, 90.32%, 86.48%, and 92.20% for As, Cd, Co, Cr, Fe, Mn, Ni, Pb, and Zn, respectively. The instrument sensitivity and reproducibility of the results and the results were reported as the averages of triple measurements.

### 2.4. Evaluation of Effluent Water Quality

#### 2.4.1. Risk Assessment Index

Humans are exposed to heavy metal contamination through oral injection, dermal absorption (skin), and inhalation (by vehicle gas exhaust). In this study, exposure assessment indices, such as non-carcinogenic risk assessment and carcinogenic risk, associated with oral ingestion and dermal absorption in the effluent samples across the studied WWTPs were considered. Among these, oral ingestion and dermal contact are the most prevalent pathways [28,29,30]. Given these two pathways, it is important to know the type of heavy metal in the effluent samples being discharged due to the potential to serve as a primary pollution source to the aquatic water body. Therefore, a model to assess non-carcinogenic hazards and carcinogenic risks to human health by heavy metals can be created via hazard quotient (HQ) or hazard index (HI) and chronic daily intake (CDI) of water. HQ in the effluent water samples across the WWTPs was calculated via Equations (1)–(3), while HI was determined by Equation (4) [31,32,33].
(1)$$ADI_{\mathrm{Oral}}=\frac{\mathrm{EC}\times \mathrm{IR}\times \mathrm{EF}\times \mathrm{ED}\;}{\mathrm{BW}\times \mathrm{AT}}$$
(2)$${\mathrm{ADI}}_{\mathrm{Dermal}}=\frac{\mathrm{EC}\times \mathrm{EF}\times \mathrm{ED}\times \mathrm{SA}\;\mathrm{ABS}\times \mathrm{ET}\times \mathrm{CF}\times K_{p\;}}{\mathrm{BW}\times \mathrm{AT}}$$
(3)$$\mathrm{HQ}=\frac{\mathrm{ADI}}{\mathrm{RfD}}$$
(4)HI=∑HQ
where ADI is the exposure dose received through the ingestion of water or dermal absorption, expressed in mg/(kg·day); EC is the average environmental concentration of trace metals (effluent water sample), expressed in mg/L; IR is the drinking water ingestion rate, considered to be 2 L/day [34]; EF is the exposure frequency, 350 day/year [35]; ED is the exposure duration, 70 years. BW is the average body weight (61.8 kg, in this study); AT is the average lifespan for non-carcinogens and carcinogens, 25,550 days; SA is the exposed skin surface area, 18,000 cm^2^; Kp is the dermal permeability constant, expressed in cm/h (1.0 × 10^−3^ for Cd, 2.0 × 10^−3^ for Cr, 4.0 × 10^−4^ for Co, 1.0 × 10^−4^ for Pb, 2.0 × 10^−4^ for Ni, and 6.0 × 10^−4^ for Zn). Additionally, ABS is the dermal absorption factor, 1.0 × 10^−3^; ET is the exposure time, 0.2 h/day; CF is the unit conversion factor for water, 1 L/1000 cm^3^; RfD is the reference dose for different heavy metals via oral ingestion and dermal contact, which was adopted from USEPA [36].The oral and dermal hazard quotient (HQ) was obtained by equaling the average oral and dermal daily intake of each heavy metal, according to the oral and dermal reference dose of each heavy metal as the case may be. The total hazard quotient for each heavy metal for each of the WWTPs was obtained by the summation of both oral and dermal HQ of the individual heavy metals. The hazard index (HI) was determined using Equation (3) and obtained by the summation of HQs for all the heavy metals. 

#### 2.4.2. Carcinogenic Risk Assessment

Exposure to a potentially carcinogenic substance, such as heavy metals, could result in cancer risk (CR), which is the incremental probability of an individual developing cancer during their lifetime. In this study, CR was evaluated in the effluent samples for the selected heavy metals with the available oral and dermal slope factor (SF) values using Equation (5), while the cancer risk index (RI) was evaluated using Equation (6). In this study, slope factors by oral intake were 0.00850 mg/kg/day, 1.7 mg/kg/day, 0.5 mg/kg/day, and 0.501 mg/kg/day for Pb, Ni, Cd, and Cr, respectively. A slope index by the dermal contact of 42.5 mg/kg/day was used for Pb and 20 mg/kg/day was available for both Ni and Fe [37]. The risk index (RI) was obtained by the addition of CR for the available SF heavy metals, as noted earlier.
(5)CR =ADI×SF
(6)RI=∑CR

## 3. Results and Discussion

### 3.1. Heavy Metal Concentrations in Influent and Effluent Samples

The levels of heavy metals in the raw influent and final effluent samples collected across the four WWTPs in this study are presented in Figure 1a,b. All the targeted heavy metals (As, Cd, Co, Cr, Fe, Mn, Ni, Pb, and Zn) were ubiquitously detected in both influent and effluent samples. The mean concentrations of all the heavy metals in the influent samples ranged from 0.122 to 1.808 mg/L, while the effluent samples have a concentration ranging from 0.118 to 0.854 mg/L. Generally, across the WWTPs, Fe was observed to be more prevalent among the investigated heavy metals in both influent (0.233–1.809 mg/L) and effluent (0.185–0.855 mg/L) samples. The levels recorded in this study were below the maximum limit of 5 mg/L in the effluent water sample, as given by USEPA [38], although it was higher than the recommended maximum concentration (RMC) [39,40] of 0.07 mg/L and 0.3 mg/L in surface water. Agoro et al. [41] reported elevated levels of Fe in influent (6.588 mg/L) and effluent (0.636 mg/L) samples from Eastern Cape, South Africa. The main route of Fe in wastewater are runoff from roofs, the wear of tires, food, large enterprises, car washes, metal surface treatment, electroplating, electronics manufacturing, organic chemicals manufacturing, iron and steel industry, mines, and quarries [42,43]. Research had shown that a high concentration of Fe in wastewater could contribute significantly to soil acidification and lead to the loss of available phosphorus and molybdenum when applied to the soil [44]. An elevated concentration of Fe can cause tissue damage and some other diseases, such as anemia and neurodegenerative conditions in humans [45]. The study by Bahiru [18] was focused on the determination of heavy metal concentrations in wastewater from Central Ethiopia. The concentrations of Fe reported were in the range of 2.89–5.13 mg/L, which is higher than what is reported in this study. Similarly, Abagale et al. [46] reported an elevated concentration of Fe (4.930–8.933 mg/L) in wastewater from car washing bays used for agriculture in the Tamale metropolis, Ghana. Cadmium and Mn were also significant in both influent and effluent samples with concentrations in the range of 0.400–0.420 mg/L and 0.400–0.410 mg/L, and 0.179–0.761 mg/L and 0.168–0.482 mg/L in the influent and effluent samples, respectively. The level of Cd reported in this study was higher, in surface water, than the RMC by WHO of 0.003 mg/L [47] and of 0.01 mg/L by DWAF [48] in effluent water, and the level of Mn was also higher than the recommended level of 0.2 mg/L in surface water [1,49] and 0.1 mg/L for aesthetic use in South Africa. These concentrations were lower than the levels of Mn (1.073–4.204 mg/L) and above the detection limit (bdl) of <0.002 mg/L for Cd as reported by Abagale et al. [46]. Agoro et al. [41] reported mean concentrations of Cd that were in the range of 0.11–0.12 mg/L in effluent samples from wastewater treatment plants in Eastern Cape South Africa. An elevated concentration of Cd (5 mg/L) was also reported by Teijon et al. [50] in the treated wastewater of the Depurbaix facility in Spain. Cd occurs naturally in soil and minerals in the form of sulfide, sulfate, carbonate, chloride, and hydroxide salts as well as in water. Industrial activity is another potential route that could lead to high concentrations of Cd in water and soil, therefore resulting in substantial Cd exposure to humans. The exposure of humans to Cd through contaminated water could lead to disturbances of important body mechanisms and may potentially result in short-term or long-term disorders [51]. The acute or chronic inhalation of Cd in industrial areas might lead to renal tubular dysfunction and lung injuries [52]. Metallic Mn is used mainly in steel production, together with cast iron and super alloys, to improve hardness, stiffness, and strength [53]. The levels of Mn in wastewater samples may substantially increase as the Mn concentration in iron-containing materials increases. Mn has the potential to induce iron deficiency in aquatic organisms, such as blue-green algae, and can lead to the inhibition of chlorophyll synthesis [54]. The concentrations of Pb and As were in the range of 0.262–0.317 mg/L and 0.266–0.299 mg/L, and 0.295–0.303 mg/L and 0.295–0.299 mg/L in the influent and effluent samples, respectively. The concentrations of Pb were found to be higher than the RMC of 0.01 mg/L and 0.015 mg/L as given by WHO and USEPA in surface water and effluent water, respectively. The level obtained in this study was lower than the value, recommended by USEPA for water reclaimed from effluents for irrigation, of 5 mg/L, but higher than 0.006 mg/L for wastewater effluents and the 0.01 mg/L standard set in South Africa for drinking water. Pb concentrations in this study were lower than the 1.98–3.11 mg/L that was reported by Bahiru [18] in wastewater samples from Ethiopia, but higher than the reported mean concentrations of 0.0153 mg/L in Kenya [55]. The major route of Pb into the environment is through metal production, cables, and pipelines, as well as paints and pesticides. It is a non-essential heavy metal and is another metal known to have damaging effects on human health [56]. Exposure to Pb can result in the alteration of the physiological functions of the body and is linked to many diseases [57,58,59]. Pb is highly toxic and capable of causing many adverse effects on the neurological, biological, and cognitive functions of humans. The concentrations of As found in the effluent samples were higher than the WHO RMC of 0.01 mg/L (10 μg/L) for As in drinking water [60] and higher than the permissible levels of 0.01 mg/L for drinking water in South Africa (SANS-241-1:2015) [61]. Additionally, they were higher than the 0.01 and 0.1 mg/L levels for groundwater and wastewater, respectively, in Poland [62]. Anthropogenic activities, such as mining, the burning of fossil fuels, medicine, electronics, pesticides, herbicides, insecticides, fertilizers, livestock as well as wood preservatives, could promote the distribution of As in the environment [63,64]. The health-related effects of As include hindering the normal metabolism of cells and causing cell death, and chronic As poisoning can lead to skin keratinization and even canceration [65]. Other prominent heavy metals were Zn and Ni with concentrations in the range of 0.211–0.367 mg/L and 0.133–0.341 mg/L, and 0.181–0.220 mg/L and 0.179–0.199 mg/L in the influent and effluent samples, respectively. The concentrations found in this study were lower than the RMC of 2 mg/L in water for Zn [1,49]. South Africa recommends ≤5 mg/L of Zn and ≤0.07 mg/L for Ni in drinking water and 2 mg/L in effluent water samples [38,66]. It should be noted that Zn toxicity in effluent sample sources is minimal and may not lead to secondary pollution sources in the receiving surface water. Zn is an essential element in the human diet in proper amounts to maintain the correct functioning of the immune system and normal brain activity, and it is critical for the proper growth and development of fetuses, although at high concentrations, it is very toxic and, therefore, may be harmful to the human body [46,67]. However, Ni concentrations call for concern due to their elevated levels in the effluent sample, which may be detrimental to aquatic and human lives. Levels of Ni in wastewater of approximately 30% have been reported by Sorme and Lagerkvist [43], which possibly originate from chemicals added in the WWTPs and released from metal cookware during cleaning [43]. The major effects of Ni on human health may include dermatitis, allergy, organ diseases, and cancer of the respiratory system [68]. Cr and Co concentrations were found to be the lowest among the heavy metals investigated in this study with concentrations in the range of 0.119–0.132 mg/L and 0.119–128 mg/L, and 0.118–0.128 mg/L and 0.115–124 mg/L in the influent and effluent samples, respectively. The tolerable level of Cr in drinking water in South Africa is ≤0.05 mg/L, while the WHO and USEPA standards for surface water and wastewater are 0.05 mg/L and 0.1 mg/L, respectively. The values obtained in this study for the influent and effluent samples were higher than the set standards, indicating that, particularly, the effluents’ interaction with the receiving body of water may pose secondary pollution and the aquatic environment may be compromised by Cr pollution. Pollution associated with the Cr found in the environment, in most cases, originates from industrial activities, such as those of the metallurgical, refractory, and chemical industries, which release a large amount of Cr into the soil, surface water, and groundwater [69]. The stable form of Cr (Cr^6+^) is a toxic pollutant due to its harmful effects on human health, with effects that range from dermal, renal, neurological, and GI diseases to the development of several cancers, including of the lungs, larynx, bladder, kidneys, testicles, bone, and thyroid [46,69]. The level of Co in the samples was above the permissible limits as stipulated by WHO (0.05 mg/L) and USEPA (0.03 mg/L) for drinking water. Co is a dispersed metal element that rarely occurs as an independent mineral in nature; therefore, it is commonly found in sulfur mines, along with other heavy metals, such as Cu, iron, lead, zinc, and other heavy metals [70,71]. Therefore, the pollution of the river’s environment is of concern as Co may be transported along the river course where water is used for various anthropogenic purposes and whose toxic effect can be manifested over time. The possible contamination with and prolonged exposure to Co may result in risks to plant growth and the survival of aquatic fauna and humans [72,73,74]. Co is also poisonous, can cause certain cancer-related diseases and, ultimately, threaten human life [71].

### 3.2. Non-Carcinogenic Risk Assessment

The quality and safety of the final effluents discharged into the receiving surface water is critical due to the potential exposure to heavy metals by aquatic and human lives. Four different WWTPs were investigated for heavy metal pollution where the influent and effluent water samples were collected and analyzed. The results of individual heavy metals reveal that substantial amounts of these heavy metals are present in the raw influent water samples and, surprisingly, the levels recorded in the effluent water samples are also significant. This is cause for concern as the effluents from these WWTPs are frequently discharged into the receiving surface water near these plants. Therefore, there is a need to know the risks to human health associated with these heavy metals as, when they are in the aquatic environment, they can jeopardize the safety of the individuals who use these surface rivers. Due to the crucial nature of the final effluent samples, non-carcinogenic and carcinogenic risk assessment studies evaluated the data generated from the effluent samples. In this study, oral ingestion and dermal contact (absorption) were considered, as they are the major route through which humans are exposed to heavy metal contamination. The hazard quotient (HQ) of individual heavy metals through oral ingestion and dermal contact as well as the hazard index (HI) for individual WWTPs were evaluated. More importantly, their HQ and HI are crucial because these WWTPs are located at different catchment areas and discharge their final effluents to different rivers; therefore, it is important to evaluate their health hazards individually. The results of HQ and HQ for oral ingestion and dermal contact are presented in Table 1, while the total HQ (THQ) is presented in Table 2. The results reveal that the HQ for oral ingestion was significantly higher for all the heavy metals compared to the HQ for dermal contact. It can be observed that oral ingestion may be a primary route through which heavy metal contaminations occur for humans and this could be due to the ingestion of contaminated foods, either directly or indirectly. Heavy metals have the potential to bioaccumulate and become biomagnified in the environment over a long time, as well as translocate in the plant and enter the fatty tissues of animals, where they can easily enter humans who consume these animals. Among the heavy metals, the THQs of As > Pb > Cr > Cd are all greater than 1. It has been reported that, if the HQ or HI > 1, it indicates that a high potential risk be present, which may likely be manifested after a long time of exposure [36,37,75,76]. The other heavy metals, such as Zn < Fe < Mn < Co < Ni, had HQ values lower than 1, which indicates that there may be no potential non-carcinogenic health risk or chronic toxicity after exposure to these heavy metals. However, although the effects may not be felt immediately, the routine assessment of these heavy metals is important. All the HI in this study were greater than the unity (>1), which occurred because As, Pb, Cr, and Cd contributed the larger percentage (As: 37.15%, Pb: 26.01%, Cr: 15.30%, and Cd: 15.22%) and may pose a serious threat to health safety due to their chronic toxicity. A similar observation was reported for As by Moloi et al. [77], who conducted a sensitivity analysis for the exposure routes in two rivers and the results showed that As was found to be the main contributor to THQ, which occurred due to the high toxicity level of As. Alves et al. [28] also reported the potential for non-cancer risk through ingestion of and bathing in contaminated waters due to the prevalence of As in Pardo River, Brazil. A high concentration of As in groundwater systems in Slovakia caused an increase in chronic and carcinogenic risk levels in a nationwide study [78]. Therefore, the non-carcinogenic risks associated with As, Pb, Cr, and Cd are considered to be medium in this study, indicating that the probable adverse health effects associated with these heavy metals are very likely or reflecting the serious potential health risks associated with the ingestion of these heavy metals. Ni was considered low (it is unlikely that any adverse health effects occur) and Zn, Fe, Mn, and Co had negligible values (no adverse health effects). Additionally, the non-carcinogenic risks due to dermal absorption for all the heavy metals investigated in this study are negligible, as the values of HQ and HI obtained for all the heavy metals were far below the unity (HQ and HI < 0.1); therefore, any adverse health effects arising from dermal adsorption are not likely to occur [79,80].

### 3.3. Carcinogenic Risk Assessment

The cancer risk (CR) and risk index (RI) for the oral ingestion of Cd, Cr, Ni, and Pb and the dermal contact of Pb, Fe, and Ni were evaluated due to the non-availability of the slope factor for the other heavy metals. The CR and IR values for the selected heavy metals are presented in Table 3. According to the values obtained in this study, the CR values for the oral ingestion of Cd, Cr, and Ni are within 10^−3^, while the CR values for Pb were found to be within the range of 7.0059 × 10^−5^–7.884 × 10^−5^. It should be noted that Cd, Cr, and Ni can be considered to be medium risk so an individual who is exposed to these heavy metals has a tendency to develop cancer, but the chances of developing cancer due to Pb exposure are very low [81]. Additionally, there is a high tendency for an individual, if ingesting these heavy metals whole orally or through any other route, to experience cancer development considering the RI values, which are less than 10^−3^. However, cancer development associated with dermal absorption is quite negligible; thereby, it does not raise any concern because the CR values for dermal contact are in the range of 1.50 × 10^−9^–6.818 × 10^−9^, 2.857 × 10^−10^–3.174 × 10^−10^, and 9.988 × 10^−11^–1.122 × 10^−10^ for Fe, Ni, and Pb, respectively. Similarly, the RI values for dermal contact across the sites was also in the range of 1.860 × 10^−9^–7.235 × 10^−9^. These RI values could be considered very low in light of the research of Haque et al. [82], indicating an extremely low risk of cancer development for an individual who was exposed to Fe, Ni, or Pb through the dermal route.

## 4. Conclusions

Human waste, such as domestic, industrial agrochemicals, and pharmaceutical waste, is a major issue when it is not adequately handled. The pollution of watercourses and canals can cause serious damage to the aquatic environment and affect vegetation, fish, and fauna negatively. Poor and inadequate sanitation systems have been considered not only a human health issue, but also an environmental threat with a high potential to instigate climate change. The influent and effluent water samples that were analyzed in this study were collected from four different wastewater treatment plants across the Durban metropolis. The status of effluent pollution by heavy metals (As, Cd, Co, Cr, Fe, Mn, Ni, Pb, and Zn) and the possible non-carcinogenic cancer index, using hazard quotient (HQ) and cancer risk (CR), were investigated in this study. The results indicate that the selected heavy metals were present in both the raw influent and final effluent samples. The mean concentrations of the heavy metals in the influent samples were in the range of 0.122–1.808 mg/L, while the effluent samples had a concentration range of 0.118–0.854 mg/L. Generally, across the WWTPs, Fe was observed to be more prevalent among the investigated heavy metals in both the influent (0.233–1.809 mg/L) and effluent (0.185–0.855 mg/L) samples. However, the levels recorded in this study were below the maximum limit of 5 mg/L in the effluent water sample (USEPA 1999), but higher than the recommended maximum concentration (RMC) of 0.07 mg/L by USEPA (2012) and 0.3 mg/L by WHO (2011) in surface water. Cr and Co concentrations were found to be the lowest among the heavy metals investigated in this study with concentrations that were in the range of 0.119–0.132 mg/L and 0.119–128 mg/L, and 0.118–0.128 mg/L and 0.115–124 mg/L in the influent and effluent samples, respectively. The non-carcinogenic risks associated with As, Pb, Cr, and Cd were considered to be medium in this study, indicating that probable adverse health effects are very likely to occur. It is unlikely that any adverse health effects occur by exposure to Ni and the values of Zn, Fe, Mn, and Co were negligible (no adverse health effects). The cancer risk evaluation showed that there is a high tendency for an individual that ingests heavy metals whole to have probable cancer development. However, the non-carcinogenic risks due to dermal absorption for all the heavy metals investigated in this study are considered negligible as the values of HQ and HI obtained for all the heavy metals due to dermal contact were far below the unity (HQ and HI < 0.1) and, thus, negligible. Therefore, any adverse health effects arising from dermal adsorption are not likely to occur and may not raise any concern in the environment.

## Figures and Tables

**Figure 1 toxics-10-00340-f001:**
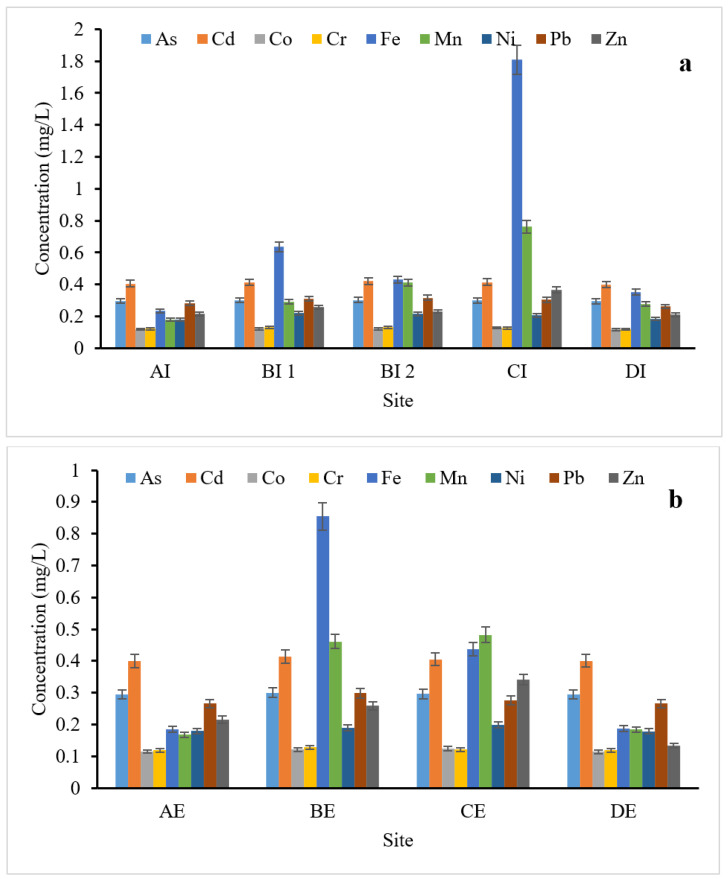
Concentrations of heavy metals: (**a**) influent sample and (**b**) effluent sample. AI, AE = Isipingo wastewater influent and effluent; BI 1, 2 BE = Southern wastewater influent and effluent; CI, CE = New Germany influent and effluent; DI, DE = Northern wastewater influent and effluent.

**Table 1 toxics-10-00340-t001:** Hazard quotient (HQ) for oral ingestion and dermal adsorption for the selected heavy metals in the effluent samples.

**Sample**		**HQ Effluent Sample (Oral Ingestion)**								
	**Metal**	**As**	**Cd**	**Co**	**Cr**	**Fe**	**Mn**	**Ni**	**Pb**	**Zn**
A E		3.046	1.241	8.93 × 10^−2^	1.234	1.91 × 10^−2^	3.72 × 10^−2^	2.78 × 10^−1^	2.064	2.23 × 10^−2^
B E		3.102	1.284	9.41 × 10^−2^	1.325	8.83 × 10^−2^	1.022 × 10^−1^	2.96 × 10^−1^	2.318	2.67 × 10^−2^
C E		3.066	1.257	9.64 × 10^−2^	1.260	4.53 × 10^−2^	1.067 × 10^−1^	3.086 × 10^−1^	2.144	3.53 × 10^−2^
D E		3.052	1.243	8.88 × 10^−2^	1.231	1.94 × 10^−2^	4.078 × 10^−2^	2.78 × 10^−1^	2.061	1.38 × 10^−2^
		**HQ Effluent Sample (Dermal Contact)**								
		**As**	**Cd**	**Co**	**Cr**	**Fe**	**Mn**	**Ni**	**Pb**	**Zn**
A E		1.57 × 10^−4^	2.55 × 10^−6^	9.18 × 10^−10^	5.08 × 10^−4^	2.10 × 10^−10^	2.79 × 10^−8^	1.36 × 10^−8^	7.08 × 10^−11^	1.03 × 10^−10^
B E		1.60 × 10^−4^	2.64 × 10^−6^	9.68 × 10^−10^	5.45 × 10^−4^	9.74 × 10^−10^	7.67 × 10^−8^	1.45 × 10^−8^	7.95 × 10^−11^	1.24 × 10^−10^
C E		1.58 × 10^−4^	2.59 × 10^−6^	9.92 × 10^−10^	5.18 × 10^−4^	4.99 × 10^−10^	8.02 × 10^−8^	1.51 × 10^−8^	7.35 × 10^−11^	1.63 × 10^−10^
D E		1.57 × 10^−4^	2.56 × 10^−6^	9.14 × 10^−10^	5.06 × 10^−4^	2.14 × 10^−10^	3.06 × 10^−8^	1.36 × 10^−8^	7.06 × 10^−11^	6.37 × 10^−11^

**Table 2 toxics-10-00340-t002:** Total hazard quotient (HQ) and hazard index (HI) for the selected heavy metals.

Sample	HQ Total									
	As	Cd	Co	Cr	Fe	Mn	Ni	Pb	Zn	HI Total
A E	3.0465	1.241	8.92 × 10^−2^	1.235	1.91 × 10^−2^	3.72 × 10^−2^	2.78 × 10^−1^	2.064	2.23 × 10^−2^	8.032
B E	3.1024	1.284	9.41 × 10^−2^	1.326	8.84 × 10^−2^	1.023 × 10^−1^	2.95 × 10^−1^	2.318	2.68 × 10^−2^	8.638
C E	3.0662	1.257	9.64 × 10^−2^	1.260	4.53 × 10^−2^	0.1.07 × 10^−1^	3.086 × 10^−1^	2.143	3.53 × 10^−2^	8.320
D E	3.0517	1.243	8.88 × 10^−2^	1.231	1.94 × 10^−2^	4.078 × 10^−2^	2.78 × 10^−1^	2.061	1.38 × 10^−2^	8.027

A I, A E = Isipingo wastewater influent and effluent; B I 1, 2 B E = Southern wastewater influent and effluent; C I, C E = New Germany influent and effluent; D I, D E = Northern wastewater influent and effluent.

**Table 3 toxics-10-00340-t003:** Cancer risk (CR) and risk index (RI) for the selected heavy metals.

Sample		CR (Oral Ingestion)					CR (Dermal Contact)			
	Metal	Cd	Cr	Ni	Pb	RI	Pb	Fe	Ni	RI
A E		6.203 × 10^−3^	1.851 × 10^−3^	9.459 × 10^−3^	7.016 × 10^−5^	1.758 × 10^−2^	9.988 × 10^−11^	1.473 × 10^−9^	2.862 × 10^−10^	1.860 × 10^−9^
B E		6.420 × 10^−3^	1.988 × 10^−3^	1.010 × 10^−2^	7.884 × 10^−5^	1.855 × 10^−2^	1.122 × 10^−10^	6.818 × 10^−9^	3.043 × 10^−10^	7.235 × 10^−9^
C E		6.287 × 10^−3^	1.888 × 10^−3^	1.050 × 10^−2^	7.288 × 10^−5^	1.874 × 10^−2^	1.038 × 10^−10^	3.491 × 10^−9^	3.174 × 10^−10^	3.912 × 10^−9^
D E		6.213 × 10^−3^	1.846 × 10^−3^	9.443 × 10^−3^	7.0059 × 10^−5^	1.757 × 10^−2^	9.974 × 10^−11^	1.50 × 10^−9^	2.857 × 10^−10^	1.884 × 10^−9^

## Data Availability

Data related to this article can be obtained from the Department of Chemical Engineering at Mangosuthu University of Technology (https://www.mut.ac.za/chemical-engineering/accessed on 7 May 2022) or upon request from the authors (adeyinka.gbadebo@mut.ac.za and BFemi@mut.ac.za).

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
