# Peer review of "Evaluating the Potential Health Risks of Selected Heavy Metals across Four Wastewater Treatment Water Works in Durban, South Africa"

_toxics, 2022, doi:10.3390/toxics10060340_

Round 1
Reviewer 1 Report
Here, Nine heavy metals; Arsenic (As), Cadmium (Cd), Cobalt (Co), Chromium (Cr), Iron (Fe), Manganese (Mn), Nickel (Ni), Lead (Pb), and Zinc (Zn) were evaluated in the influent and effluent water samples from four wastewater treatments across Durban metropolis, KwaZulu-Natal, South Africa. The results indicated that the mean concentrations of all the heavy metals in the influent samples ranged from 0.122 to 1.808 mg/L while effluent samples have a concentration range of 0.118 to 0.854 mg/L. Iron was found to have the highest concentrations and the concentration of Co was the least across the wastewater treatment plants. The levels found for most of the heavy metals in this study were found above the recommended maximum concentrations in surface water and effluent water as stipulated by the World Health Organization, United State Environmental Protection Agency, Food and Agriculture Organization, and Department of Water Affairs and Forestry of South Africa. According to the toxicity effect due to non-carcinogenic risk, As, Pb, Cr, and Cd are considered medium in this study indicating probable adverse health risk is very likely to occur.
It is well written and interesting article which needs the following corrections before being accepted:
1. The English composition requires some improvements. The authors should proofread the manuscript carefully to minimize grammatical and bibliographic errors.
2. There are several articles available on the similar topic. So explain the novelty in the introduction part.
3. The authors mentioned that due to non-carcinogenic risk, As, Pb, Cr, and Cd are considered medium in this study indicating probable adverse health risk is very likely to occur. While all these metals are very toxic and carcinogenic. How they considered these non-carcinogenic?
4. The authors should explain the abbreviated words as a footnote of all tables.
5. The quality of all figures must be improved.
6. It would be better if the authors discuss in brief in introduction part about the methods like adsorption, ion exchange and phytoremediation for metal removal. Following work or other similar work may be followed: Ionics 21 (8), 2237-2245, 2015; Desalination and Water Treatment 54, 2883-2890, 2015; Environmental chemistry letters 16 (4), 1339-1359, 2018
Author Response
Reply to reviewer’s comments
The authors would like to appreciate the reviewer for the in-depth evaluation of this manuscript “Evaluating Potential Health Risk of Selected Heavy Metals across Four-Wastewater Treatment Water Works within Dur-ban, South Africa". The general overview and the insight provided by the reviewer have greatly improved the quality of the manuscript. All the suggestions and comments provided by the reviewer have been carefully attended to and we believe that the revised manuscript should now be suitable for the Toxics.
Comments and Suggestions for Authors
Here, Nine heavy metals; Arsenic (As), Cadmium (Cd), Cobalt (Co), Chromium (Cr), Iron (Fe), Manganese (Mn), Nickel (Ni), Lead (Pb), and Zinc (Zn) were evaluated in the influent and effluent water samples from four wastewater treatments across Durban metropolis, KwaZulu-Natal, South Africa. The results indicated that the mean concentrations of all the heavy metals in the influent samples ranged from 0.122 to 1.808 mg/L while effluent samples have a concentration range of 0.118 to 0.854 mg/L. Iron was found to have the highest concentrations and the concentration of Co was the least across the wastewater treatment plants. The levels found for most of the heavy metals in this study were found above the recommended maximum concentrations in surface water and effluent water as stipulated by the World Health Organization, United State Environmental Protection Agency, Food and Agriculture Organization, and Department of Water Affairs and Forestry of South Africa. According to the toxicity effect due to non-carcinogenic risk, As, Pb, Cr, and Cd are considered medium in this study indicating probable adverse health risk is very likely to occur.
It is well written and interesting article which needs the following corrections before being accepted:
- The English composition requires some improvements. The authors should proofread the manuscript carefully to minimize grammatical and bibliographic errors.
Response: The English composition has been checked and re-evaluated accordingly.
- There are several articles available on the similar topic. So explain the novelty in the introduction part.
Response: The importance of this study has been highlighted in the introductory part of the manuscript as suggested. “In recent times, the availability and supply of potable water in South Africa have witnessed a dramatic decline. The inadequate supply of safe water due to the water stress in the country has increased the demand for the surface water where people could be exposed to avoidable diseases and other related health issues and ecological imbalance as a result of pollution, which may arise from anthropogenic sources. Also, the recent emergence of the Covid-19 pandemic has not only affected people’s behaviour but its overall effects have contributed to the pollution source such as heavy metals in the aquatic environment which emanating from the non-pharmaceutical means of preventing the spread of the virus. The levels of pollutants such as heavy metals in the raw sewage that frequently end up in the surface water body require constant and routine evaluation to safeguard the aquatic environment and health of humans, therefore, prompted this study”.
- The authors mentioned that due to non-carcinogenic risk, As, Pb, Cr, and Cd are considered medium in this study indicating probable adverse health risk is very likely to occur. While all these metals are very toxic and carcinogenic. How they considered these non-carcinogenic?
Response: The reviewer's observation is appreciated. Yes, these heavy metals (As, Pb, Cr, and Cd) are known to be toxic and carcinogenic. The insight given by the authors in this study was based on values of hazard quotient (HQ) values for these heavy metals for non-carcinogenic prediction. The HQ values for these heavy metals were found to be greater than 1 (HQ > 1) which indicated that, high potential risk due to other non-cancer related issues such as kidney and heart problems, dermatologic, neurologic, reproductive, and genotoxic effects which may occur due to chronic exposure.
- The authors should explain the abbreviated words as a footnote of all tables.
Response: The abbreviated words have been explained as a footnote of the tables as suggested
- The quality of all figures must be improved.
Response: Authors appreciate your view on the use of graphical representation of the data with Microsoft excel. The use of other software like origin 9 to improve the quality of the figures would have been explored but currently, we require an update on this which hopefully be sorted out as soon as possible. We would be glad to explore this alternative for our next data presentation. We hope the reviewer will understand our frustrations and be so kind as to consider the only alternative we have at the moment.
- It would be better if the authors discuss in brief in introduction part about the methods like adsorption, ion exchange and phytoremediation for metal removal. Following work or other similar work may be followed: Ionics 21 (8), 2237-2245, 2015; Desalination and Water Treatment 54, 2883-2890, 2015; Environmental chemistry letters 16 (4), 1339-1359, 2018
Response: These have been incorporated into the main manuscript as suggested.

Reviewer 2 Report
The article by Bakare et al is timely however major improvements need to be made before I can accept this article
Anytime heavy metals in water systems comes up people are going to pay attention to it, and I believe this article will be cited a lot.
I believe the introduction with the cited articles is good but can be improved. Recently with all the disposable facial masks this will be another source of water pollution, as many masks are non-biodegradable. I believe the following article should be cited:
Bussan, D. D., Snaychuk, L., Bartzas, G., & Douvris, C. (2022). Quantification of trace elements in surgical and KN95 face masks widely used during the SARS-COVID-19 pandemic. Science of The Total Environment, 814, 151924.
That article quantifies trace elements in facial masks which these masks end up in the water systems.
For the quality control were there any reference materials used such as ERM-CA713 to make sure that the instrument was performing correctly? If so what was the percent recovery? This is important as arsenic can be a difficult metal to measure by ICP-OES.
I would like to see a table inserted that includes the ICP-OES parameters that were used in this experiment. This way another researcher could reproduce these same results under the same conditions.
In line 285 the authors mention that the permissible levels for arsenic are 10 mg/L, this is extremely high, the recommendation is 10 µg/L, when you look at the authors results, they have for figure a and b site AE has arsenic levels at 300 ppb, which is 30 times higher than the permissible levels by WHO. I have personally seen labs get false positive numbers for arsenic when using ICP-OES so I would caution that a reference material is needed to verify such high numbers.
Before I accept this manuscript, the authors need to consider the aforementioned points as some of the points are very serious and crucial to this paper.
Author Response
Reply to reviewer’s comments
The authors would like to appreciate the reviewer for the in-depth evaluation of this manuscript “Evaluating Potential Health Risk of Selected Heavy Metals across Four-Wastewater Treatment Water Works within Dur-ban, South Africa". The general overview and the insight provided by the reviewer have greatly improved the quality of the manuscript. All the suggestions and comments provided by the reviewer have been carefully attended to and we believe that the revised manuscript should now be suitable for the Toxics.
Comments and Suggestions for Authors
The article by Bakare et al is timely however major improvements need to be made before I can accept this article
Anytime heavy metals in water systems comes up people are going to pay attention to it, and I believe this article will be cited a lot.
I believe the introduction with the cited articles is good but can be improved. Recently with all the disposable facial masks this will be another source of water pollution, as many masks are non-biodegradable. I believe the following article should be cited:
Bussan, D. D., Snaychuk, L., Bartzas, G., & Douvris, C. (2022). Quantification of trace elements in surgical and KN95 face masks widely used during the SARS-COVID-19 pandemic. Science of The Total Environment, 814, 151924.
That article quantifies trace elements in facial masks which these masks end up in the water systems.
Response: The authors appreciate the constructive insight and useful suggestions of the reviewer. The suggested study by Bussan et al., (2022) has been incorporated in the introductory section accordingly.
For the quality control were there any reference materials used such as ERM-CA713 to make sure that the instrument was performing correctly? If so what was the percent recovery? This is important as arsenic can be a difficult metal to measure by ICP-OES.
I would like to see a table inserted that includes the ICP-OES parameters that were used in this experiment. This way another researcher could reproduce these same results under the same conditions.
Response: The instrument performance was routinely monitored daily before any analyses. The instrument sensitivity and other important parameters such as align view, detector calibration, UV/Vis wavelength calibration; precision, spectral resolution, detection limits, and stability were as well monitored on the instrument. The ERM-CA713 was not used for this study; this would be considered forthwith in our next study, as the study is still ongoing to consider the best remediation approach in removing these heavy metals in the wastewater systems to safeguard the health of the human and ecological system in the environment. Although, other important quality control measures were carefully performed and most importantly blank samples were simultaneously analyzed with the real samples to know if there were interferences arising from other sources. The blank samples were found to contain insignificant amounts of the analyzed heavy metals which in most cases were found below the detection limits for the respective heavy metals.
The instrument parameters have been provided in the main manuscript as suggested. “The ICP-OES parameters are; the argon gas was used set at 415.82 kPa, power was at 1.19 Kw, plasma was at 9.99 and auxiliary 0.60 and carrier flow was at 0.70 with low purge flow. The rotation pump was set at 20 revolutions per minute (20 r.p.m) with speed, temperature set at 37.99℃ while CCD temperature was operated at -14.99℃ and vacuum level was at 1.5 Pa”.
In line 285 the authors mention that the permissible levels for arsenic are 10 mg/L, this is extremely high, the recommendation is 10 µg/L, when you look at the authors results, they have for figure a and b site AE has arsenic levels at 300 ppb, which is 30 times higher than the permissible levels by WHO. I have personally seen labs get false positive numbers for arsenic when using ICP-OES so I would caution that a reference material is needed to verify such high numbers.
Response: The authors appreciated this important observation by the reviewer this is such a significant observation and a great contribution to this study. The value quoted (10 mg/L) with the unit was such an oversight, the unit was wrongly written, it was supposed to be 10 µg/L and this has been corrected accordingly in the main manuscript. The value of As reported (0.3 mg/L) in this study was higher than the permissible levels of 0.01 mg/L (10 µg/L) in drinking water as recommended by the WHO and the South African regulatory bodies. The values reported in this study were for effluent samples to be discharged into the receiving surface water, which could undergo further dilution and partition into the soil and sediment compartment in the aquatic environment whereby a substantial amount may not be available to the aquatic life. Therefore, the real concentrations in the receiving rivers are worthy of investigation periodically. Although, there is an ongoing study on the remediation approach for the removal of heavy metals from the wastewater which we hope would be beneficial in removing these heavy metal contaminants from the wastewater system before their release into the water body.
Before I accept this manuscript, the authors need to consider the aforementioned points as some of the points are very serious and crucial to this paper.
Response: Authors would like to appreciate the reviewer once again for the good insight and the professional touches given to this manuscript, which have, improve the quality of this manuscript. We hope the issues raised have been carefully addressed and the reviewer would be pleased with the author's responses and be glad to consider this manuscript for publication in the Toxics.

Round 2
Reviewer 2 Report
I would like to thank the authors for making the necessary corrections. I am willing to accept the article since the corrections have been addressed.
Author Response
Comment
Again, make sure to provide analytical performence for the measured metals. What was the recovery for each metal? This is important certain elements such as arsenic can be a difficult to measure by ICP-OES.
Response: The issues raised have been attended to in the main manuscript accordingly. Now read as; “The instrument response factor (R2) for each heavy metal was found to range from 0.998 to R2 > 0.999. The limit of detection and quantifications were evaluated using three (3) and ten (10) times the standard deviation of the blank with a slope of the regression line respectively. The detection limits are 0.066 µg/mL (As), 0.0294 µg/mL (Cd), 0.0583 µg/mL (Co), 0.0281 µg/mL (Cr), 0.0595 µg/mL (Fe), 0.0220 µg/mL (Mn), 0.0473 µg/mL (Ni), 0.0249 µg/mL (Pb) and 0.0636 µg/mL (Zn). While the limit of quantification are 0.219 µg/mL, 0.098 µg/mL, 0.194 µg/mL, 0.094 µg/mL, 0.198 µg/mL, 0.073 µg/mL, 0.157 µg/mL, 0.083 µg/mL and 0.212 µg/mL for As, Cd, Co, Cr, Fe, Mn, Ni, Pb and Zn respectively. The recovery study was done by spiking a known concentration of each heavy metal in deionized water, followed the same procedure as it was performed for the real samples. The percentage recovery was found to be 82.93%, 84.98%, 91.91%, 86.87%, 91.30%, 94.85%, 90.32%, 86.48% and 92.20% for As, Cd, Co, Cr, Fe, Mn, Ni, Pb and Zn as the case may be. The instrument sensitivity and reproducibility of the results and the results were reported as the averages of triple measurements”.
